# Influence of Microstructure on Synchrotron X-ray Diffraction Lattice Strain Measurement Uncertainty

**Chris A. Simpson** [1,*]**, David M. Knowles** [2,3] **and Mahmoud Mostafavi** [3,*]

1  Perpetual Robotics, T Block, UWE, Frenchay Campus, Coldharbour Lane, Stoke Gifford, Bristol BS16 1QY, UK
2  Sir Henry Royce Institute for Advanced Materials, The University of Manchester, Royce Hub Building, Manchester M13 9PL, UK; david.knowles@bristol.ac.uk
3  Department of Mechanical Engineering, University of Bristol, Queen's Building, University Walk, Bristol BS8 1TR, UK
*  Correspondence: c.a.simpson01@gmail.com (C.A.S.); m.mostafavi@bristol.ac.uk (M.M.); Tel.: +44-(0)117-331-5717 (C.A.S.); +44-(0)7969-2020 (M.M.)

**Abstract:** Accurate residual lattice strain measurements are highly dependent upon the precision of the diffraction peak location and the underlying microstructure suitability. The suitability of the microstructure is related to the requirement for valid powder diffraction sampling statistics and the associated number of appropriately orientated illuminated. In this work, these two sources of uncertainty are separated, and a method given for both the quantification of errors associated with insufficient grain sampling statistics and minimization of the total lattice strain measurement uncertainty. It is possible to reduce the total lattice strain measurement uncertainty by leveraging diffraction peak measurements made at multiple azimuthal angles. Lattice strain measurement data acquired during eight synchrotron X-ray diffraction experiments, monochromatic and energy dispersive, has been assessed as per this approach, with microstructural suitability being seen to dominate total measurement uncertainty when the number of illuminated grains was <$10^6$. More than half of the studied experimental data fell into this category, with a severe underestimation of total strain measurement uncertainty being possible when microstructural suitability is not considered. To achieve a strain measurement uncertainty under $10^{-4}$, approximately $3 \times 10^5$ grains must be within the sampled gauge volume, with this value varying with the multiplicity of the family of lattice planes under study. Where additional azimuthally arrayed data are available an in-plane lattice strain tensor can be extracted. This improves overall strain measurement accuracy and an uncertainty under $10^{-4}$ can then be achieved with just $4 \times 10^4$ grains.

**Keywords:** synchrotron X-ray diffraction; uncertainty; strain; grain size; monochromatic; energy dispersive

## 1. Introduction

Accurate evaluation of the development and distribution of residual lattice strain (and the associated quantification of residual stress) is a key consideration in structural integrity assessment procedures, such as R6 [1] and BS7910 [2,3]. Experimental measurement of lattice strain is typically achieved through either X-ray or neutron diffraction and the associated determination of Bragg diffraction peak locations, with these measurements routinely being made at large scale synchrotron X-ray or neutron facilities. In both cases accurate strain measurement is typically dominated by (a) the accurate location of the diffraction peak center and (b) microstructure suitability. While the former has been covered in great depth by Withers et al. [4], the latter has not been systematically explored to the best of authors' knowledge.

In this case, microstructure suitability refers to the interplay between the microstructure and the experimental setup, with this being most broadly determined by the sampled gauge volume and size of the material's crystallographic units (nominally the grain size).

More specifically, a suitable microstructure would be one in which the number of appropriately orientated grains illuminated by the incoming beam are enough for a precise measure of the bulk strain in the measured location and orientation. Requisite strain measurement accuracy will clearly vary with experimental need, but is typically of the order of $10^{-4}$ to $10^{-5}$ [4]. This level of accuracy is reliant upon precise diffraction peak definition and high microstructure suitability, these rely upon good counting and powder diffraction sampling statistics, respectively.

While both described sources of uncertainty are present irrespective of radiation source, the balance between them is, however, likely to shift with technique. Neutron diffraction (ND) can reasonably be considered a flux (photon count) limited acquisition method. In this case, a relatively large gauge volume is required (typically $> 10$ mm$^3$ [5–7]), which will help satisfy the microstructure suitability criterion, with uncertainty therefore [8] being dominated by counting statistics and the associated precision of the diffraction peak location. The opposite is more likely, although not always guaranteed, in synchrotron X-ray diffraction, where beam flux is orders of magnitude higher and measurement gauge volumes are often orders of magnitude lower ($<<1$ mm$^3$). In either case, it is critical that the dominant factor is identified and controlled for prior to the start of any experimental session and, furthermore, that true measurement uncertainty is precisely captured and not underestimated.

There have been several attempts to study variance and measurement uncertainty in ND measurements. The most notable examples of which is through the European Network on neutron techniques standardization for structural integrity task group 1 (NET TG1) [9] and task group 4 (NET TG4) [10]. These were round robin measurement exercises carried out on a single weld bead applied on a 316L(N) plate. While uncertainty in the peak center location was quoted as giving rise to a residual stress uncertainty of 20 MPa, a true measurement uncertainty (taken from the inter-laboratory variance in measured stress) approximately twice this was observed. The authors suggest that this additional uncertainty could be linked to a grain size effect, although no grain size was explicitly defined.

### 1.1. Peak Location Precision

It is common to approximate diffraction peaks with a Gaussian or Lorentzian profile (or some combination thereof), defined by $X$, the peak center location, $u_x$, the standard deviation of the peak and $H$, the peak height. This profile is applied onto a background function, which is typically a constant term, $B$ [4]. The associated uncertainty in the center of a Gaussian peak with either a low signal to background ratio (H:B > 10) has been analytically evaluated by Withers et al. [4] and shown to be inversely proportional to the square root of the integrated intensity, such that:

$$\Delta_X \cong \frac{u_x}{I^{1/2}} \tag{1}$$

where $\Delta_X$ is the uncertainty in the peak position and $I$ is the total integrated intensity. This relationship is supported by empirical data provided by both Withers et al. [4] and Webster and Kang [11]. There is, however, a significant deviation from this relationship and the associated expected precision with increasing background signal and, more specifically, where $H : B < 10$. Nevertheless, in both cases, uncertainty decreases as counting time and total integrated intensity increases. Given the very high brilliance and flux in synchrotron diffraction, it is often trivial (and not overly costly in terms of time) to produce high intensity peaks with low peak fitting errors.

### 1.2. Microstructure Suitability

When a small number of grains are sampled grain-scale strain heterogeneity is apparent and pseudo-strains are introduced through the uneven distribution of grains throughout the gauge volume. The latter of these effects is associated with a variation in the effective sample-to-detector distance [12]. The uncertainty associated with powder diffraction sam-

pling statistics is linked to the number of sampled grains and also to the degree of internal crystallographic misorientation within those grain, often referred to as mosaicity [13]. With increased mosaicity the number of diffracting units within a grain increases, with this internal misorientation itself increasing as a function of deformation and plasticity. As a lattice is deformed the uncertainty associated with the grain sampling statistics would be expected to decrease.

The lattice strain uncertainty associated with the microstructure suitability is also determined by the multiplicity of the studied diffraction peak, with the multiplicity being the number of symmetry equivalent reflections for a given combination of crystal structure and lattice family. For close-packed fcc and bcc materials, i.e., those regularly the subject of SXRD and ND residual strain measurements, commonly studied lattice plane families, {hkl}, have multiplicities ranging from 6 to 24. Selecting a diffraction peak associated with a family of planes with a higher multiplicity is equivalent to sampling a refined grain structure (or increasing the sampled gauge volume). Note that destructive interference limits the number of {hkl} families that produce diffraction peaks, and the high energy beam typically precludes the study of lattice families with large plane spacings.

It is difficult to quantify material mosaicity a priori and it more practical to simply consider the number of grains illuminated by the X-ray (or indeed neutron) beam, $N_i$, or a multiplicity normalized equivalent $\overline{N_i} = N_i \times M/10$, where $M$ is the multiplicity of the lattice plane family and dividing by 10 scales the values into a sensible range. Hutchings et al. [12] suggest that a grain size in the tens of microns should limit this source of measurement uncertainty for a gauge volume of 1 mm$^3$, with tens to hundreds of thousands of grains lying within that gauge volume. The authors also suggest a more qualitative approach, which involves looking at how spotty or continuous that diffraction pattern is or assessing the intensity variance with respect to azimuthal angle.

## 2. Methodology

### 2.1. Systematic Analysis of SXRD Experimental Error

A broad study of measurement uncertainty has been carried out on previously published experimental work [14–19], with SXRD data from a total of 8 experiments being systematically analyzed. These experiments ran across 7 synchrotron beamtime sessions and are detailed in Table 1. A grain size range between 0.5 and 100 μm is covered, with each experiment focusing on a different alloy or material system. In Table 1 there is a brief description of each experiment, alongside the gauge volume and grain size, the combination of which allows the number of grains illuminated to be calculated. The number of illuminated grains, $N_i$, ranges across four orders of magnitude, from $10^4$ to >$10^7$ grains. The precise details for each experiment to be found within the associated experiments. Note that both monochromatic, or angular dispersive, and energy dispersive data have been considered with the methodology noted in Table 1. A full description of both methods and SXRD strain measurement in general can be found in detail elsewhere (e.g., [20]) but briefly, in monochromatic X-ray diffraction, an area detector is typically used to collect the Debye-Scherrer cones produced using a fixed energy/wavelength ($\lambda$) beam, with the Bragg angle, $2\theta$, associated with diffraction peaks varying with lattice plane spacing, $d$, according to the well-known Bragg's Law:

$$2d \, \sin\left(\frac{2\theta}{2}\right) = n\lambda \tag{2}$$

With energy dispersive X-ray diffraction (EDXRD) a fixed Bragg angle is defined and a polychromatic beam is used to illuminate the sample, with photons being collected on an energy sensitive detector(s). In this case the energy at which the diffracted peaks lie is a function of lattice spacing.

**Table 1.** A list of the experiments analyzed as part of this work with the associated uncertainty in the measured lattice strain for varying lattice plane families, in this case referenced by their multiplicity, *M*.

| ID | Material | Mode | D (µm) | V (µm³) | $N_i$ | $\Delta\varepsilon$ | | | | $\Delta\varepsilon_X$ | | | | Description of Measurements | Reference |
|---|---|---|---|---|---|---|---|---|---|---|---|---|---|---|---|
| | | | | | | *M* = 6 | *M* = 8 | *M* = 12 | *M* = 24 | *M* = 6 | *M* = 8 | *M* = 12 | *M* = 24 | | |
| OL | Bainitic Steel | ED | 5 | $50 \times 50 \times 4000$ | $7.0 \times 10^4$ | $2.1 \times 10^{-4}$ | - | $1.7 \times 10^{-4}$ | $1.7 \times 10^{-4}$ | $1.2 \times 10^{-4}$ | - | $7.6 \times 10^{-5}$ | $5.5 \times 10^{-5}$ | Crack tip stress mapping to quantify and separate fatigue overload mechanisms | (Simpson et al., 2018) |
| UFG | Ultra-Fine Grain Ni | Mono | 0.5 | $50 \times 50 \times 1200$ | $3.0 \times 10^7$ | $6.9 \times 10^{-4}$ | $3.2 \times 10^{-5}$ | $8.0 \times 10^{-5}$ | $5.7 \times 10^{-5}$ | $5.2 \times 10^{-5}$ | $5.0 \times 10^{-5}$ | $3.8 \times 10^{-5}$ | $3.5 \times 10^{-5}$ | Crack tip stress mapping to study effect of anisotropic microstructure on overload crack growth | (Zhang et al., 2019) |
| SR | 316H | Mono | 130 | $500 \times 500 \times 5000$ | $9.0 \times 10^3$ | $3.8 \times 10^{-4}$ | $3.3 \times 10^{-4}$ | $3.1 \times 10^{-4}$ | $2.3 \times 10^{-4}$ | $3.5 \times 10^{-5}$ | $3.9 \times 10^{-5}$ | $2.8 \times 10^{-5}$ | $2.3 \times 10^{-5}$ | Intergranular strain development measurements made during high temperature stress relaxation | (Mamun et al., 2019) |
| ROL | AISI 52100 | ED | 15 | $150 \times 150 \times 3200$ | $4.1 \times 10^4$ | $2.3 \times 10^{-4}$ | - | - | $1.7 \times 10^{-4}$ | $1.9 \times 10^{-4}$ | - | - | $1.0 \times 10^{-4}$ | Stroboscopic analysis of contact stress development (and crack initiation) in roller bearings | (Reid et al., 2019) |
| NPV | SA508 Grade 4N | Mono | 4 | $200 \times 200 \times 2000$ | $2.4 \times 10^6$ | $8.6 \times 10^{-5}$ | - | $6.0 \times 10^{-5}$ | $3.1 \times 10^{-5}$ | $3.7 \times 10^{-5}$ | - | $2.4 \times 10^{-5}$ | $2.7 \times 10^{-5}$ | High speed crack tip stress tracking to study effects of thermal shock in nuclear pressure vessels | (Oliver et al., 2018) |
| EBW | 316L | ED | 25 | $1000 \times 1000 \times 3200$ | $3.9 \times 10^5$ | - | - | $1.2 \times 10^{-4}$ | $8.8 \times 10^{-5}$ | - | - | $1.6 \times 10^{-5}$ | $1.3 \times 10^{-5}$ | Residual stress measurements in electron beam welded steel plates | (Mokhtarishirazabad et al., 2019) |
| FCAW | ASTM A131 Grade DH36 | ED | 20 | $500 \times 500 \times 3200$ | $1.2 \times 10^5$ | $1.9 \times 10^{-4}$ | - | $1.6 \times 10^{-4}$ | $1.5 \times 10^{-4}$ | $2.6 \times 10^{-5}$ | - | $2.1 \times 10^{-5}$ | $2.3 \times 10^{-5}$ | Residual stress measurements in flux core arc welded steel plates | Unpublished |
| P91 | P91 Steel | Mono | 20 | $500 \times 500 \times 5000$ | $3.0 \times 10^5$ | $1.1 \times 10^{-4}$ | - | $1.0 \times 10^{-4}$ | $4.5 \times 10^{-4}$ | $1.5 \times 10^{-5}$ | - | $3.4 \times 10^{-5}$ | $1.6 \times 10^{-5}$ | Intergranular strain development measurements made during high temperature stress relaxation | Unpublished |

In monochromatic X-ray diffraction (using a 2D area detector), the initial processing step requires the azimuthal integration of Debye-Scherrer rings, resulting in a set of azimuthally spaced 1D line profiles. This procedure is typically well integrated into the beamline workflow and is best done as part of the acquisition pipeline. For instance, the data analysis DAWN [21,22]'s WorkbeNch, software package has the option to run azimuthal integration and output integrated profiles in convenient, multi-dimensional arrays. The ESRF have also invested significant resource into the development of ESRF's Python Fast Azimuthal Integration (PyFAI [1]), which also offers this functionality and is particularly sophisticated and feature rich [23]. Note that in energy dispersive X-ray diffraction, the raw data are already in this 1D form (albeit with respect to energy rather than angle) and no initial processing step is required.

For the monochromatic data assessed in this work (UFG, SR, MMC, NPY, P91), the integration step was completed using DAWN, with the diffraction rings being separated and integrated across 10° slices. An example of the separation and integration of diffraction rings associated with ultra-fine grain Ni (UFG) can be seen in Figure 1. Integrating across 10° azimuthal slices nominally results in 36 1D line profiles. The notable exception to this is for the 316H stress relaxation experiment, SR [18,24] and the associated, previously unpublished, stress relaxation data acquired for a high temperature ferritic steel, P91. In both cases, approximately half the outgoing Debye-Scherrer cone was blocked by auxiliary heating equipment, with the number of azimuthal bins being reduced in proportion to this, i.e., just 18 slices were analyzed.

Once the data are in the form shown in Figure 1b, all subsequent analysis was completed using the pyXe strain analysis software [25]. pyXe [25] is an in-house software developed by the authors which has been described and used elsewhere in detail. For completeness, the process followed by pyXe is briefly described in detail in Section 2.2. The software has been used previously to analyze the data from a number of synchrotron experiments (e.g., see [14–17,24]) results of which were possible compared with widely used software such as Fit2D [23]. While the code was evaluated against other results to the best of authors' ability, it is difficult to carry out a full evaluation of the code without significant resources thus care should be taken when using it.

## 2.2. Lattice Strain Analysis

User defined scripting and analysis of diffraction peaks is most optimally and sensibly applied to the assessment of 1D diffraction profiles rather than to raw 2D Debye-Scherrer rings. The steps required to efficiently analyze large volumes of 1D line profile data (and to duly convert to strain) are not currently well provided for at synchrotron facilities. In this series of experiments analysis of 1D line profiles was carried out using pyXe [25], which is a strain analysis software developed using Python's SciPy ecosystem (NumPy, SciPy, matplotlib, etc.). It is aimed at reducing bottlenecks in the strain analysis pipeline, most notably for use in synchrotron experiments, where the volume and velocity of incoming data is typically high. During large scale facility experiments it is key that analysis and visualization are carried out on-the-fly, or as near to this as possible, to allow beamline users to make informed decisions about their experiments. The pyXe analysis pipeline is shown in Figure 2.

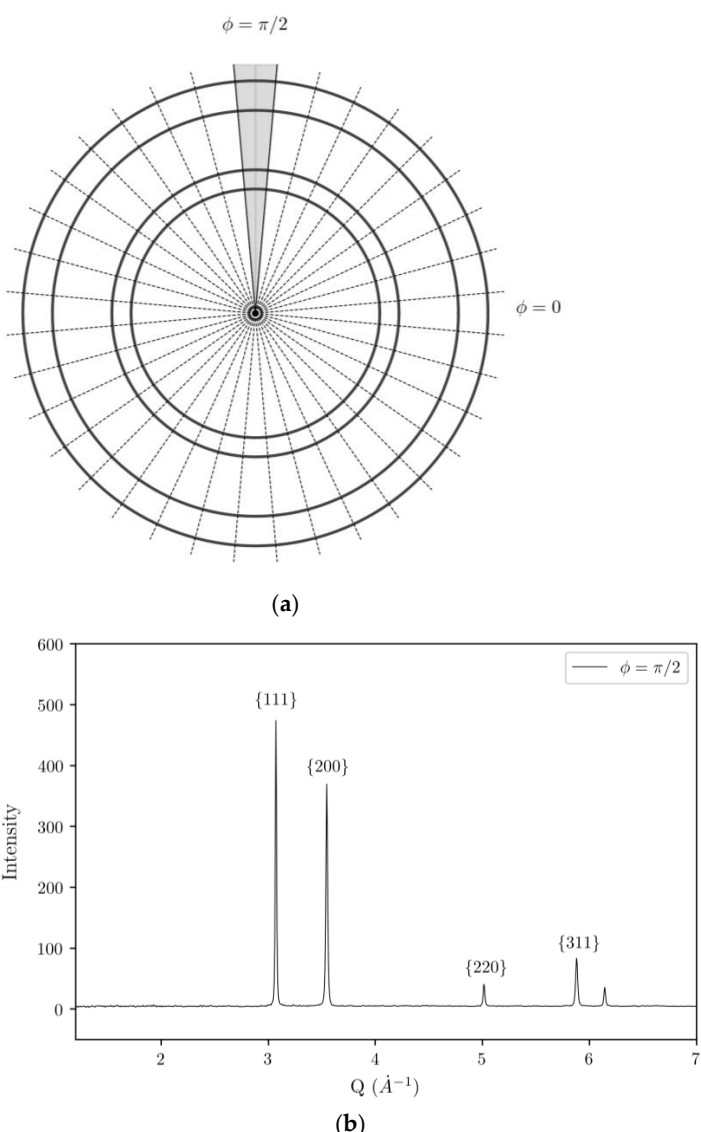

(a)

(b)

**Figure 1.** Schematic showing (**a**) the azimuthal slices, with the highlighted region delimiting the range integrated across to produce the 1D line profile shown in (**b**). The line profile given in (**b**) is associated with ultra-fine grain Ni [14].

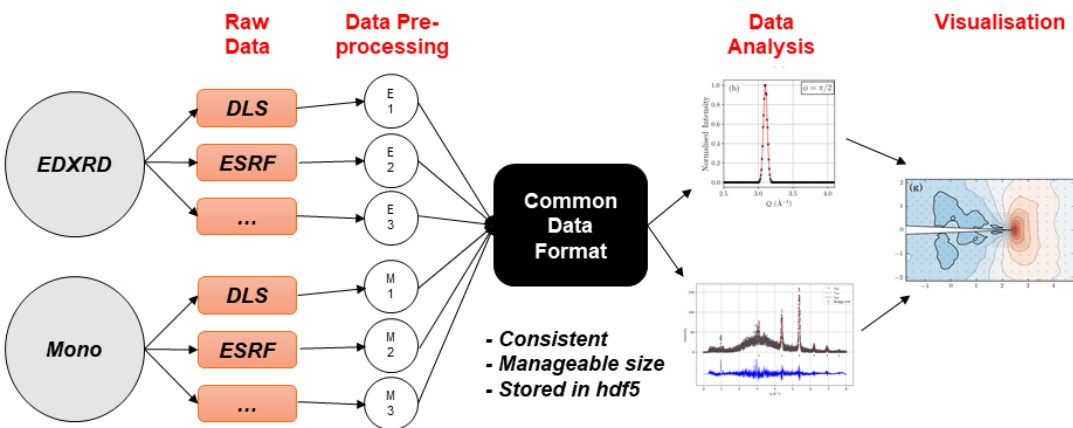

**Figure 2.** The strain analysis workflow used in pyXe.

The format in which the data are served to synchrotron beamline users varies with acquisition mode, i.e., monochromatic vs. energy dispersive, experimental facility, and beamline, and even with the experimental session (for a nominally identical setup). For this reason, a pre-processing step was used to convert the data from each of the experiments detailed in Table 1 to a common data format for consistent, convenient analysis. Key to this transformation is that the angular or energy axis is redefined with respect to the scattering vector $Q$, such that [20]:

$$Q = \frac{2\pi}{d} = \frac{4\pi \, E \, \sin\left(\frac{2\theta}{2}\right)}{hc}. \tag{3}$$

where $E$ is the energy in eV, $h$ is Planck's constant, and $c$ is the speed of light. The conversion to $Q$ allows for a consistent analysis pipeline irrespective of input data format.

*2.3. D Line Profile Analysis*

Non-linear least-squares regression was used to fit a Gaussian profile to the acquired diffraction peaks, with a five-parameter fit being employed. The peak height, $H$, peak center, $X$, and standard deviation, $u_x$, were all refined, with the background being taken to be a linear function rather than a constant value. A higher order background fit was seen to degrade the ability of the least square regression to converge.

A Poisson weighting was applied to the refinement; if a weighting system is not implemented then widening the analysis range tends to unrealistically decrease the uncertainty [4]. An unweighted fit assumes variance of all data points are equal, which is increasingly unrealistic as you move further from the peak center (and the intensity tends towards the background). While the weighted fit uncertainty is independent of fitting range (above $7 - 8u_x$), the processing time is not. Increasing the analysis window from $5u_x$ to $20u_x$ increased processing time by a factor of 3. A fit range of $\sim 7 - 8u_x$ is recommended by Withers et al. [4] as this is sufficient for minimum weighted uncertainties in the key diffraction peak parameters. An analysis window width of approximately $8u_x$ was therefore employed for all data assessed in this study.

The peaks were converted to strain by comparison to a stress free equivalent (a subject covered in great detail in [26]). In the analysis presented in this work, a stress-free lattice spacing or parameter that varies with detector or azimuthal angle has been utilized. This is not to account for a physical variation in the stress-free parameter/spacing but rather to account for, and cancel out, errors in detector alignment or setup. This approach is strongly recommended as errors introduced by incorrectly identifying the precise beam center, can lead to large to the introduction of large pseudo-strains. For example, at a sample to detector distance of 1.2 m and a Bragg angle, $2\theta = 5°$, a center misalignment of 0.1 mm (just over 0.5 pixel for a Pilatus 2M detector), results in pseudo-strains of $1 \times 10^{-3}$, which equates to a pseudo-stress of around 200 MPa for a steel alloy. When using an azimuthally varying stress-free lattice spacing, the lattice strain at a given azimuthal angle, $\varepsilon_{(\varphi, hkl)}$, is calculated according to the following relationship [20]:

$$\varepsilon'_{ij(hkl)} = Q\varepsilon_{ij}Q^T \tag{4}$$

where $Q_{(\varphi, hkl)}$ is the lattice plane specific scattering vector at the same azimuthal angle and $Q_{0\,(\varphi, hkl)}$ is its stress free equivalent.

Strain Tensor Extraction

A key aspect of the analysis used in this work is the calculation and subsequent use of the in-plane lattice strain tensor, $\varepsilon_{ij(hkl)}$, which for brevity will be referred to as $\varepsilon_{ij}$. That is to say that rather than solely looking at the 1D line profile associated with the strain orientation of interest (typically $0°$ and/or $90°$), an in-plane strain tensor is first calculated from the full set of azimuthally arrayed data. The stress state does not change as function of azimuthal angle, rather the individual components of the tensor vary with rotation relative to the global coordinate system, such that at any position around the azimuthal

the transformed in-plane lattice strain tensor, $\varepsilon'_{ij}$, can be found according to the following relationship given in Equation (5).

$$\varepsilon'_{ij(hkl)} = Q\varepsilon_{ij}Q^T \tag{5}$$

where

$$Q = \begin{bmatrix} \cos 2\varphi & \sin 2\varphi \\ -\sin 2\varphi & \cos 2\varphi \end{bmatrix}. \tag{6}$$

Each position around the azimuth represents a distinct (but related) realization of the in-plane lattice strain tensor and the materials' underlying stress state. Strain varies sinusoidally around the azimuth and by using the full set of azimuthally arrayed data it is possible to achieve a precise refinement of this relationship, and thereby the in-plane lattice strain tensor.

The in-plane lattice strain tensor can, of course, then be used to calculate the strain at the angle of interest, with utilization of the extra data reducing the total lattice strain measurement uncertainty relative to a single peak, single slice approach. Furthermore, the dispersion of measured lattice strain around the in-place lattice strain tensor also gave a key indication of the total strain uncertainty. This process and associated measurement accuracy improvements are discussed and quantified in Section 3.1.

*2.4. Measurement Gauge Volume and Illuminated Grains*

Given that the number of illuminated grains is potentially a controlling consideration for achieving satisfactory lattice strain measurement precision, it is important to understand the associated geometry of the gauge volume, $V$, and the grain size, $D$, of the material to be studied. In the monochromatic diffraction experiments covered in this work, depth resolved measurements were not made, which is to say the outgoing beam was not defined (e.g., with the conical slit system) [27]. As such, the gauge length is simply the thickness of the part being measured, $z$, with the remaining gauge dimensions being defined by the slit size $h_1 \times h_2$, where in most cases $h = h_1 = h_2$, such that $V = zh^2$. Knowing the grain size, the number of illuminated grains can then be calculated:

$$N_i = \frac{6V}{\pi D^3} \tag{7}$$

The precise gauge dimensions are less clear in energy dispersive diffraction, with the gauge length varying as a function of the incoming slit dimension. Examples of energy dispersive gauge cross sections with respect to slit size have been calculated according to Rowles et al. [28], and are presented in Figure 3a. With a sample to detector distance of 1 m, the gauge volume is effectively a parallelogram, with a fixed base, $b = 3.2$ mm. So, while the tip-to-tip length of the gauge volume, $l$, increases with increasing slit size, the actual gauge volume does not so that $V = bh^2$, $V \neq lh^2$. Incorrectly assuming the gauge volume scales with $l$ leads to a severe overestimation of gauge volume, which is shown in Figure 3b. This is significant as an overestimation of the gauge volume leads to an unrealistic estimation of the number of grains sampled, potentially leading to an underestimation of expected lattice strain uncertainty.

In the experiments detailed in Table 1 the gauge volume ranges from 0.003 to 1.25 mm$^3$, with 4 experiments having $V \leq 0.125$ mm$^3$ and a mean gauge volume of approximately 0.5 mm$^3$. The associated number of illuminated grains in each experiment varied from 9000 to 30,000,000.

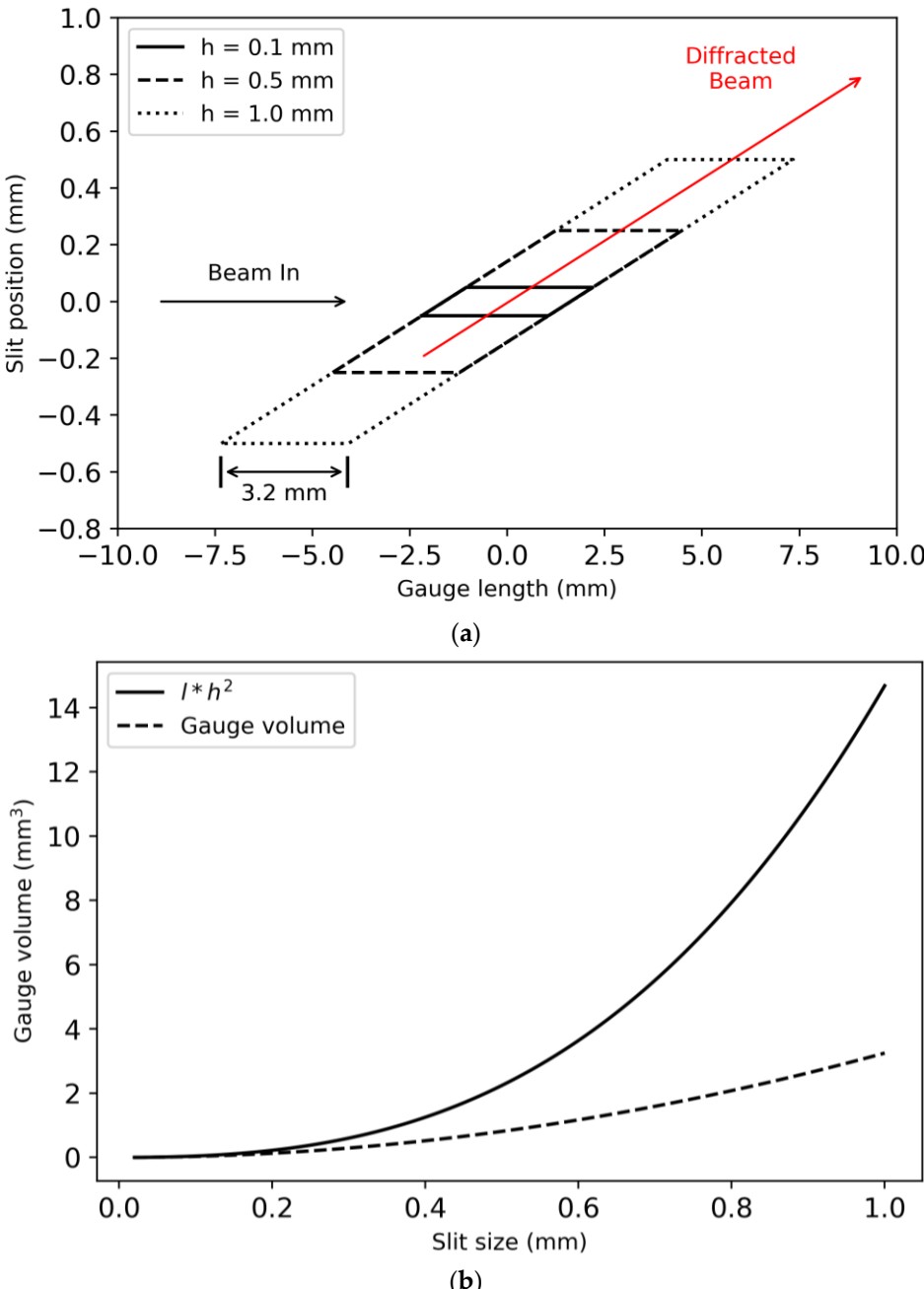

**Figure 3.** EDXRD gauge volume with respect to incoming slit size (I12:JEEP) as per Rowles et al. [29]. (**a**) Two dimensional depiction of the gauge volume (**b**) change of gauge volume as a function of slit size.

## 3. Results

### 3.1. Quantifying SXRD Measurement Uncertainty

An example of a 1D line profile is shown in Figure 4a; these data were acquired as part of the *FCAW* experiment (see Table 1), with the diffraction peak associated with the {211} family of planes being assessed for an azimuthal slice located at an angle $\varphi = 90°$. The fit of a Gaussian profile to this diffraction peak is shown in Figure 4b, with the peak center being found to lie at $Q = 5.365 \; A^{-1}$ which corresponds to a lattice spacing, $d = 1.171 \; \mathring{A}$. The uncertainty and, more specifically, the standard deviation in the peak center location, $\Delta_X$

(in this case $\Delta_X$ is the uncertainty with respect to the scattering vector, $\Delta_Q$), can be used to define the lattice strain uncertainty related to the peak center precision, $\Delta\varepsilon_X$, such that [20]:

$$\Delta\varepsilon_X = \left(\frac{Q_0}{Q_0 - \Delta_Q}\right) - 1. \tag{8}$$

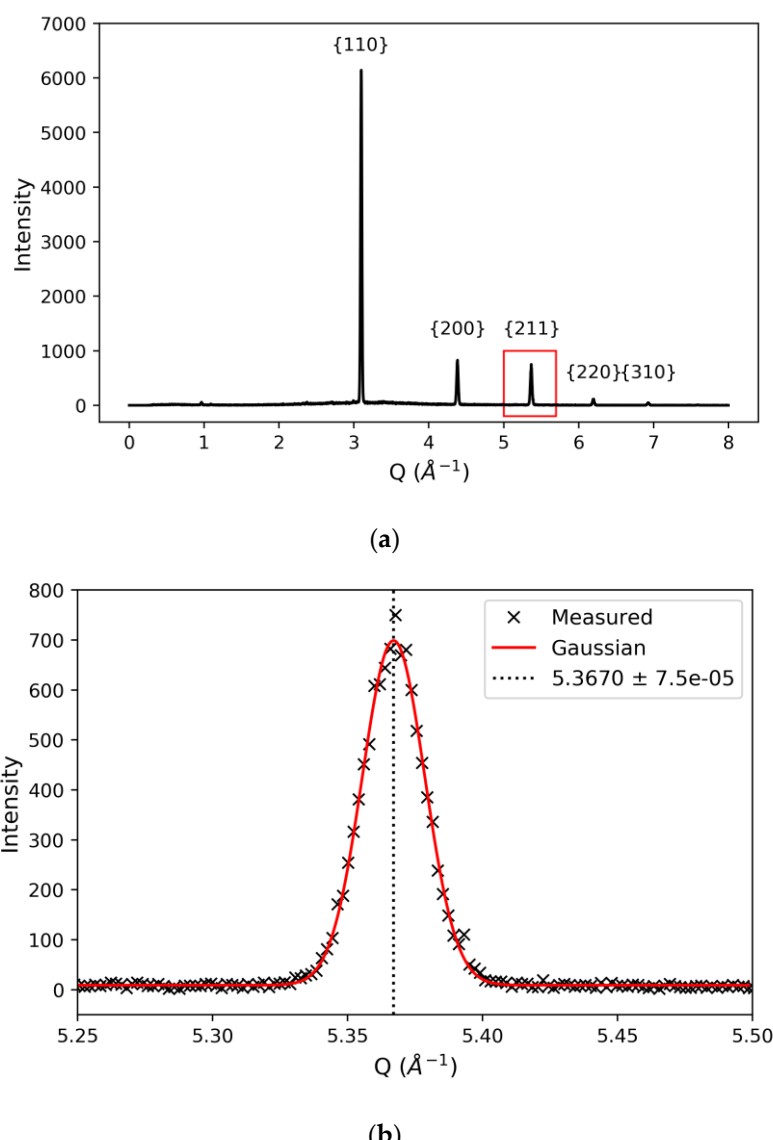

**(a)**

**(b)**

**Figure 4.** (**a**) 1D line profile associated with the FCAW residual stress data, with the diffraction peak associated with the {211} family of planes highlighted and (**b**) the fit of a Gaussian profile to this peak.

The profile depicted in Figure 4 has a particularly high to background ratio, which results in a precise measure of the peak center location and low level uncertainty, such that $\Delta_X = 6.9 \times 10^{-5} A^{-1}$. This introduces an associated uncertainty in lattice strain of $\Delta\varepsilon_X = 1.3 \times 10^{-5}$. While this certainly represents one source of error, it is not clear that this is an upper bound on the true measurement error. Indeed, the NeT-TG4 round robin neutron diffraction measurements highlighted much larger uncertainties in weld residual stresses than were predicted from the uncertainties in the peak center locations [10].

In Figure 5a, the peak fitting procedure and associated strain calculation has been applied to the full set of azimuthally arrayed data. As noted in Section 3.2.1, the strain is seen to vary sinusoidally with respect to azimuthal angle, with the lattice strain defined by the in-plane lattice strain tensor and the transformation relationship described in

Equation (5). Figure 5a highlights both the measured strains and the best fit approximation of $\varepsilon_{ij}$. The difference between the in-plane lattice strain tensor and individual measurement points is given in Figure 5b; the distribution of these errors allows us to interrogate the total lattice strain measurement uncertainty, $\Delta\varepsilon$, which is given by the root mean squared difference between the in-plane lattice strain tensor and individual measures of strain. The deviation of the strain away from the in-plane lattice strain tensor will reduce as the peak fit error improves and the number of grains increase. In Figure 5b, $\Delta\varepsilon$ is compared against the distribution of errors that would be expected from the uncertainty in the peak center location. When $\Delta\varepsilon \cong \Delta\varepsilon_X$ the total measurement uncertainty is dominated by the uncertainty associated with the location of the peak center. This is not the case for the data given in Figure 5b, where $\Delta\varepsilon = 1.5 \times 10^{-4}$, which is order of magnitude larger than $\Delta\varepsilon_X$. In this case, total lattice strain measurement uncertainty is dominated by other factors, most likely related to the microstructural suitability.

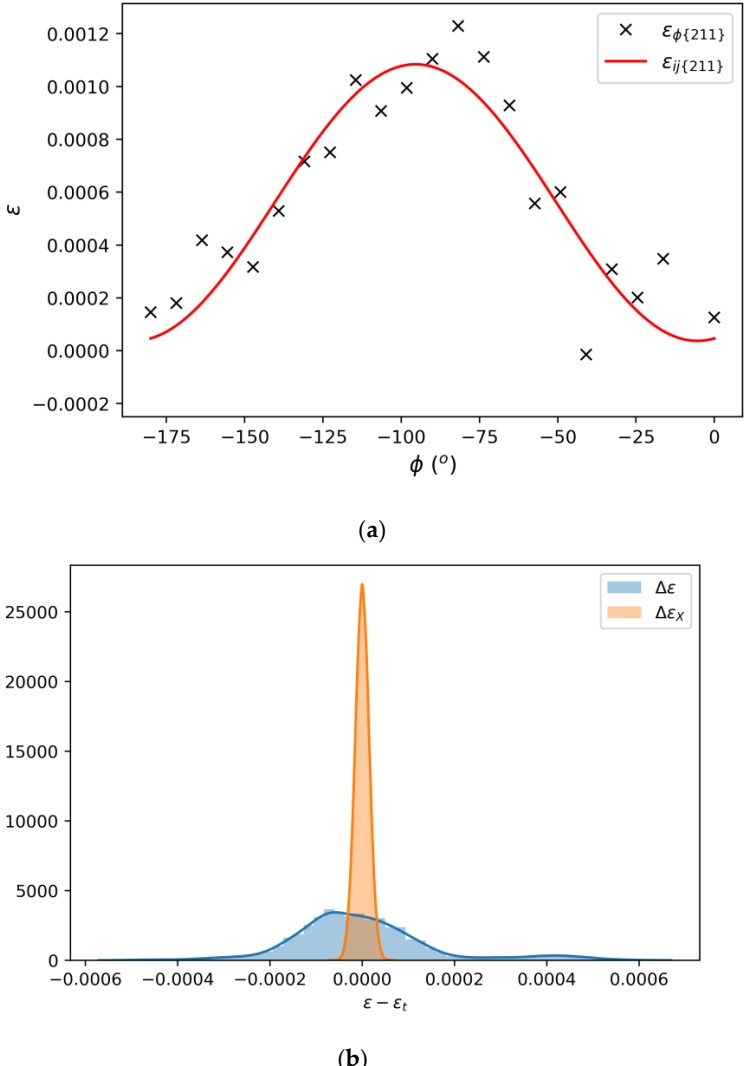

(a)

(b)

**Figure 5.** (a) Measured strain with respect to azimuthal angle for a single measurement point compared against the associated in-plane strain tensor calculated from the azimuthally varying data. (b) The distribution of differences between the measured strain and those calculated from the in-plane tensor—the data in (b) are for all 203 measurement points acquired for a single residual strain mapping scan (FCAW) and are compared against the expected distribution of errors associated with the uncertainty in the peak center location.

While $\Delta\varepsilon$ is representative of the combined measurement error for a single azimuthal slice, when using $\varepsilon_{ij}$ to define the strain at a given angle, the quoted error should be the uncertainty in the in-plane lattice strain tensor, $\Delta\varepsilon_{ij}$. In the example presented in Figure 5, the uncertainty in all three tensor components is $\Delta\varepsilon_{ij} = 6.2 \times 10^{-5}$, which represents more than a two-fold reduction in uncertainty relative to using a single azimuthal slice. In the following sections, the described approach is applied systematically to the set of data and experiments described in Table 1.

### 3.2. Systematic Analysis of Measurement Uncertainty

An extensive body of experimental data has been systematically analyzed, with measurement uncertainties ($\Delta\varepsilon$, $\Delta\varepsilon_{ij}$, $\Delta\varepsilon_X$) being extracted from each of the 8 experiments listed in Table 1. For each of those experiments, measurement uncertainties have been calculated and detailed for multiple families of lattice planes, with a total of 24 distinct crystallographic conditions being studied across a diverse range of experimental configurations.

#### 3.2.1. Inter-Experimental Variation in Strain Uncertainty for a Fixed Multiplicity

The variation in measurement uncertainties are shown in Figure 6 for lattice plane families with multiplicities of 24 (i.e., {311} for fcc materials and {211} for bcc materials), with the data for other lattice families being detailed in Section 3.2.2. Across this body of experiments, the total lattice strain measurement uncertainty ranges from $\Delta\varepsilon = 3.5 \times 10^{-5}$ to $\Delta\varepsilon = 2.2 \times 10^{-4}$. This is, on average approximately 3 times greater than the uncertainty associated with the precision of the diffraction peak location. ($1.3 \times 10^{-5}$ to $1.0 \times 10^{-4}$). In short, $\Delta\varepsilon_X$ is a non-conservative and inconsistent measure of the total lattice strain measurement uncertainty. There was only 1 experiment (*NPV*) where $\Delta\varepsilon_X \cong \Delta\varepsilon$ and this work being on a fine-grained material, with a relatively large number of grains being illuminated ($D = 5$ μm, $N_i = 10^6$). More generally, there is a clear relationship between the number of illuminated grains and total strain measurement error, with this systematically decreasing while $N_i < 500,000$ ($N_i^{-1} > 2 \times 10^{-6}$).

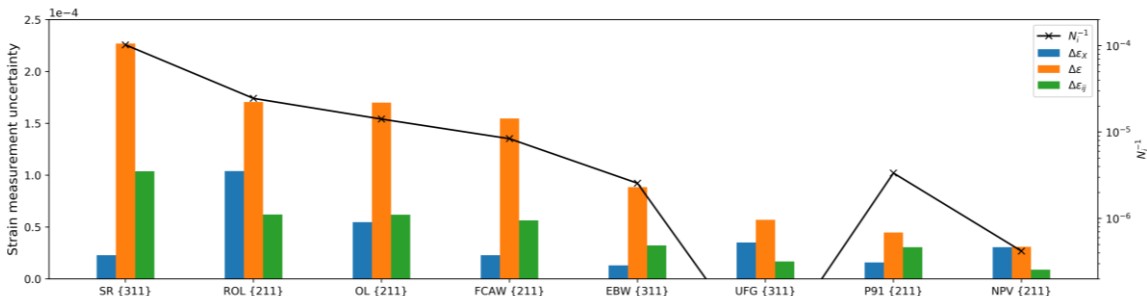

**Figure 6.** Lattice strain measurement for the experiments listed in Table 1, for lattice plane families with a multiplicity of 24. The number of sampled grains are given in inverse. Uncertainty is compared against the number of illuminated grains.

The total strain measurement uncertainty was improved by over a factor of 3 when the full set of azimuthally arrayed data was utilized and the in-plane lattice strain tensor calculated.

#### 3.2.2. Effect of Multiplicity on Strain Uncertainty

The impact of lattice plane family selection and the associated multiplicity on SXRD measurement uncertainty is shown in Figure 7. The presented data are for a stress relaxation (SR) experiment [18] on 316H stainless steel. The grain size in these samples was large (around 130 μm) and the number of illuminated grains was low at approximately $10^4$. As expected, there is an increase in measurement precision with an increase in multiplicity. The total lattice strain measurement error reduces from $3.8 \times 10^{-4}$ to $2.2 \times 10^{-4}$ as multiplicity increases from 6 to 24, i.e., the {200} versus the {311}. In terms of an equivalent stress measurement uncertainty, $\Delta\sigma$, this is an improvement from approximately $\Delta\sigma = 75$ MPa

to $\Delta\sigma = 44$ MPa. In all cases, calculating the in-plane lattice strain tensor reduces the uncertainty by more than a factor of 2, bringing the uncertainty in stress for the {311} family of lattice planes down to $\Delta\sigma = 20$ MPa. Note that as multiplicity increases, $\Delta\varepsilon_X$ is also decreasing, albeit only marginally. This is to be expected as increased multiplicity is also associated with an increase in peak intensity; peak location uncertainty is inversely related to the square root of the integrated intensity [20].

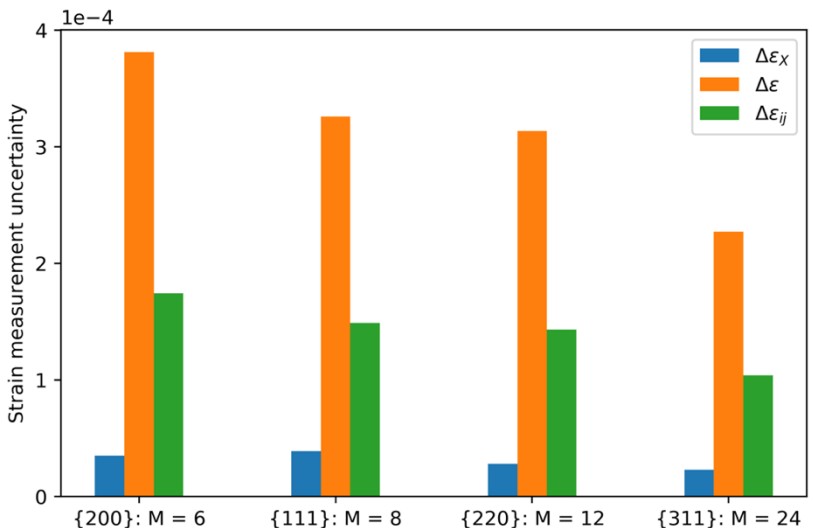

**Figure 7.** The effect of multiplicity on SXRD strain measurement uncertainty for experiment *SR*, a stress relaxation carried out on 316H stainless steel.

### 3.2.3. Number of Multiplicity Normalized Grains vs. Strain Uncertainty

The strain measurement uncertainty for each of the 8 analyzed experiments were detailed for between 3 and 4 lattice plane families; their multiplicities ranged between 6 and 24. Analyzing multiple lattice plane families for each experimental data set allows measurement error to be assessed across a far wider range of crystallographic conditions, effectively increasing the scope and extent of the measurements. Given that increased multiplicity is proportional to an increase in the number of illuminated grains, the measurement uncertainty for each experiment and lattice plane family combination has been related to the multiplicity normalized number of sampled grains, $\overline{N_i}$. The relationship between $\overline{N_i}$ and measurement uncertainty can be seen in Figure 8, with a systematic decrease in $\Delta\varepsilon$ and $\Delta\varepsilon_{ij}$ being observed while $\overline{N_i} < 7 \times 10^5$, with lattice strain uncertainty plateauing beyond this point.

Again, the uncertainty in the in-plane lattice strain tensor scaled with the $\Delta\varepsilon$, with $\Delta\varepsilon_{ij} \cong \Delta\varepsilon/3$ for all monochromatic and energy dispersive data. On a more granular level, the improvement in accuracy is dependent on the amount of data used to fit the in-plane strain tensor; for a 23-element EDXRD detector this improvement was by a factor of 2.7, for the full-ring monochromatic experiments (caked into 36 slices), the improvement was by a factor of 3.5. For monochromatic experiments where only a partial ring is captured (*SR*, *P91*), this uncertainty improvement drops to just over a factor of 2.

$\Delta\varepsilon_X$ is not correlated with the true measurement uncertainty across the range of experiments and microstructural conditions sampled here. Even where $\overline{N_i} > 7 \times 10^5$ and grain sampling effects are no longer dominating, there is no convincing correlation between $\Delta\varepsilon_X$ and $\Delta\varepsilon$, which is due to the highly precise peak centre positioning and low value of $\Delta\varepsilon_X$ across all of these experiments.

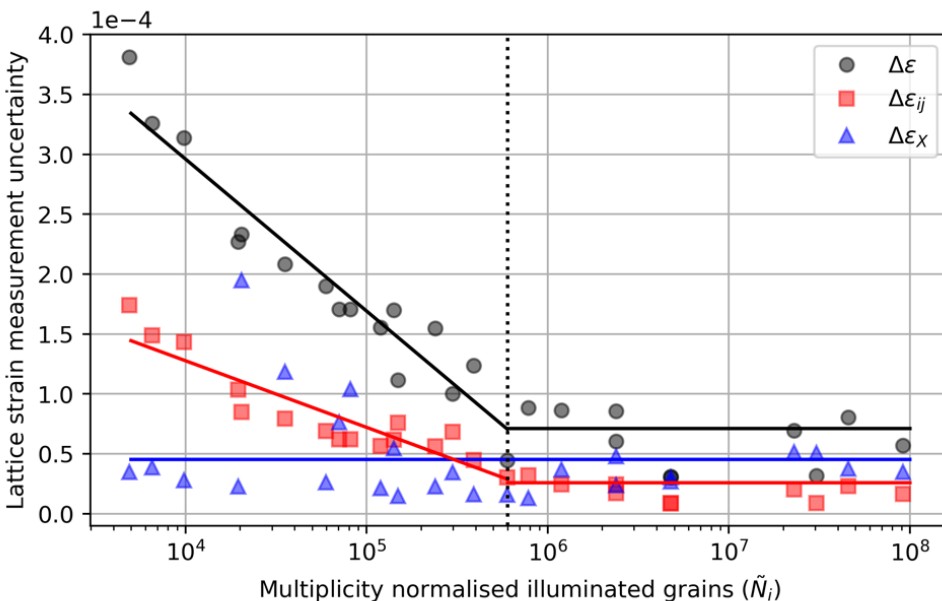

**Figure 8.** Multiplicity normalized grains against measurement error.

## 4. Discussion

This work has clearly demonstrated that the uncertainty associated with the peak fitting and the location of the peak center should be considered a lower bound, unconservative estimate for the total SXRD strain measurement uncertainty. This is consistent with the findings from the NeT-TG1 and TG4 round robin ND residual stress measurements, where fitting errors of 20 MPa were quoted, but the actual uncertainty was approximately double this [10,29] $\Delta\varepsilon_X$ does not capture variability in the detector and beamline setup or errors in the $d_0$ measurement and calibration, nor does it identify scenarios in which there are insufficient grains within the gauge volume. In the latter scenario in particular, very large errors can be masked by very low peak center uncertainties. This is particularly problematic if the $\Delta\varepsilon_X$ is being used as an initial data acceptability criterion during a SXRD diffraction experiment. Invalid uncertainty estimates can lead to bad data being collected. The peak fit error is only expected to contribute when $\Delta\varepsilon_X > 10^{-4}$; for the majority of SXRD strain measurements it is trivial to achieve peak location precision that is better than this (in the experiments studied in this work, the mean value of $\Delta\varepsilon_X$ was under $5 \times 10^{-5}$).

Fitting an in-plane lattice strain tensor to azimuthally arrayed strain data offers two key advantages: (a) it produces a better estimate of the true strain measurement uncertainty for a single azimuthal slice, capturing the effects of detector, beamline, grain size, etc., and (b) it allows for the reduction in that uncertainty. The best estimate of the actual measurement uncertainty has been quantified here as the RMSE of the in-plane tensor to the measured strain. This represents the dispersion and variance of the strain measurements around the best estimate of the in-plane strain state. If $\Delta\varepsilon_X$ captured the total measurement uncertainty, then the RMSE would be expected to approximate to $\Delta\varepsilon_X$. This was, however, not the case and furthermore there was no relationship between $\Delta\varepsilon_X$ and this best estimate of true measurement error. This ratio varied from 2.5 to almost 30, depending on the setup and experiment and reaffirms the idea that the fitting error does not represent a good or even consistent estimate for the measurement error.

There is a close correlation between $\Delta\varepsilon$ and $\Delta\varepsilon_{ij}$, the latter of which varies with the number of azimuthal points that are leveraged. While this work has used a fixed 10° azimuthal slice for the monochromatic data, there will be an optimal number of azimuthal slices for a given monochromatic dataset. This will be a tradeoff between the reduction in both the $H : B$ ratio and number of illuminated grains for a given azimuthal slice and the improved accuracy of the least-square refinement of the in-plane tensor with an increased number of azimuthal bins.

The lowest errors were achieved in the UFG experiment (an overload fatigue experiment on ultra-fine grain Ni); this is perhaps unsurprising given the ultra-fine grain nature of the material, with a very large number of grains being illuminated ($N_i > 10^7$, $D = 0.5$ μm). While the measurement precision was very high, it is important to recognize that the final accuracy of the strain data were not as good as this might suggest. In this experiment measurements were being made around a crack tip and, as such, the strain gradient was high compared to the gauge area. The upshot of this is significant smoothing of the strain field, particularly close to the crack-tip. While this is not unexpected, it is a good case study for the kind of challenges that are associated with quantifying uncertainty in diffraction experiments. It is critical that measurement uncertainty is both quantified and reduced, and that additional sources of potential error are considered and included where appropriate. Quoting unrealistically low (or high) errors is of significant concern as these can propagate through into assessments procedures.

Grain size effects are a much more important consideration in synchrotron X-ray diffraction experiments compared to their neutron diffraction equivalent. This is due to what is typically a significant difference in the measurement gauge volume. In ND measurements the gauge volume is typically of the order of 64 mm$^3$ (4 mm × 4 mm × 4 mm) [5]. In contrast an SXRD experiment generally utilizes a small incoming slit size, so as to define the gauge volume to less than 0.5 mm$^3$ (e.g., 5 mm × 0.3 mm × 0.3 mm), with this potentially dropping below of 0.01 mm$^3$ in many reasonable combinations of sample and slit size (see Table 1). The difference in setup represents what is typically a two to three order of magnitude reduction in gauge volume and, therefore, a two to three order of magnitude reduction in the number of grains illuminated. With a 50 μm grain size, a 64 mm$^3$ gauge volume illuminates 10$^6$ grains, which is a regime in which microstructural effects are no longer dominant. For a typical SXRD gauge volume of 0.5 mm$^3$, only 6000 grains would be illuminated for that same microstructure, and grain size effects should be expected to dominate. Note that these calculations are only applicable for an un-textured microstructure. Where texture is significant (e.g., in weld fusion zones), the number of appropriately orientated grains may be much lower than would nominally be expected—it is, of course, vital that this is taken into consideration. The increased significance of peak location precision in ND is also likely due to the requirement for long exposure times; while it is generally trivial to achieve high peak intensities (and high signal-to-background ratios) in SXRD, this is not always the case in ND, as the associated neutron flux is orders of magnitude lower than for SXRD.

## 5. Conclusions

- Total measurement uncertainty in synchrotron X-ray diffraction is badly characterized by the uncertainty in the diffraction peak center location. Peak location precision is only likely to significantly contribute to measurement uncertainty when it is the grain size compared to gauge volume is large and the number of sampled grains is small;
- The number of illuminated grains if a far better indicator of measurement precision, with the total SXRD measurement uncertainty being strongly correlated with the number of illuminated grains and the associated multiplicity of the studied family of lattice planes;
- For a single peak fit of an fcc or bcc material the {311} or {211} lattice family should be studied where practicable. These families are not significantly affected by intergranular strains, and are also associated with high multiplicities, which will reduce measurement uncertainty in many grain limited SXRD experiments;
- To achieve acceptable measurement precision (defined as less than $10^{-4}$ in strain) across a single azimuthal slice more than 300,000 grains should be sampled. Leveraging additional azimuthally arrayed data and calculating an in-plane strain tensor can reduce this requirement by more than an order of magnitude, with similar precision being achieved with just 40,000 grains;

- Characterizing the in-plane lattice strain tensor not only allows experimenters to reduce measurement uncertainty but it also provides them with a tool with which to more robustly quantify that error.

**Author Contributions:** Conceptualization, C.A.S.; methodology, C.A.S.; software, C.A.S.; formal analysis, C.A.S.; writing—original draft preparation, C.A.S.; writing—review and editing, M.M. and D.M.K.; supervision, M.M. and D.M.K.; funding acquisition, M.M. and D.M.K. All authors have read and agreed to the published version of the manuscript.

**Funding:** This research received no external funding from EDF as part of High Temperature Centre, European Commission as part of H2020 ATLAS+ Project and EPSRC (grant number: EP/R020108/1).

**Data Availability Statement:** Data will be provided upon request sent to corresponding author (C.A.S.).

**Acknowledgments:** We thank Diamond Light Source for access to beamline I12:JEEP (EE12205, EE16647, EE18468, EE18836, EE16096) that contributed to the results presented here.

**Conflicts of Interest:** The authors declare no conflict of interest.

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
