# Peer review of "Influence of Microstructure on Synchrotron X-ray Diffraction Lattice Strain Measurement Uncertainty"

_metals, doi:10.3390/met11050774_

Round 1

Reviewer 1 Report

Comments are given in the attached pdf.

Reviewer 2 Report

This article details assessed the lattice strain measurement data acquired from eight synchrotron X-ray diffraction experiments to examine the effects of microstructure suitability on the accurate strain measurements in materials. This paper appears well put together, but there are a few minor issues.

1. Please provide references for all your equations.

2. There are some minor typos in the text that need to be fixed, such as there being two number 5s for the Conclusions section header.

Reviewer 3 Report

Dear authors,

thank you very much for submitting the manuscript to the Metals journal.

The manuscript deals with the quantification of errors associated with insufficient grain sampling statistics and minimization of the total lattice strain measurement uncertainty that occur during measurements of residual stresses. The authors studied various parameters (peak location precision, number of irradiated grains, multiplicity of the studied family of lattice planes) and their influence on measurement uncertainties. From this point of view, the study has a huge practical impact. I also very appreciate the recommendation of choosing the FCC and BCC {311} and {211} diffracting plane families associated with high multiplicities in order to reduce the uncertainties. It is the very useful information that can be used by XRD users. The introduction provides sufficient theoretical background. All references used in the manuscript are relevant. Methods used during the experiment are described adequately and in detail. So, the experiment could be easily repeated. The structure of experiment is defined in logical sequence. Results are clearly presented and supports achieved results. Based on above, I do not have any additional questions, or suggestions for improvements. I found just one typo in entire manuscript. Please, correct it.

Line 295 - there is a typo, the author used twice a preposition „in“ at the beginning of the sentence.

So, based on above mentioned, I recommend to publish the article after the minor revision.

Best regards
